# Mechanical Properties and Microstructure of DMLS Ti6Al4V Alloy Dedicated to Biomedical Applications

**DOI:** 10.3390/ma12010176

**Published:** 2019-01-07

**Authors:** Żaneta Anna Mierzejewska, Radovan Hudák, Jarosław Sidun

**Affiliations:** 1Faculty of Mechanical Engineering, Bialystok University of Technology, ul. Wiejska 45c, 15-351 Białystok, Poland; j.sidun@pb.edu.pl; 2Department of Biomedical Engineering and Measurements, Technical University of Košice, ul. Letná 1/9, 04200 Košice, Slovakia; radovan.hudak@tuke.sk

**Keywords:** selective laser melting, direct metal laser sintering, porous biomaterials, titanium alloys, mechanical properties

## Abstract

The aim of this work was to investigate the microstructure and mechanical properties of samples produced by direct metal laser sintering (DMLS) with varied laser beam speed before and after heat treatment. Optical analysis of as-built samples revealed microstructure built of martensite needles and columnar grains, growing epitaxially towards the built direction. External and internal pores, un-melted or semi-melted powder particles and inclusions in the examined samples were also observed. The strength and Young’s modulus of the DMLS samples before heat treatment was higher than for cast and forged samples; however, the elongation at break for vertical and horizontal orientation was lower than required for biomedical implants. After heat treatment, the hardness of the samples decreased, which is associated with the disappearance of boundary effect and martensite decomposition to lamellar mixture of α and β, and the anisotropic behaviour of the material also disappears. Ultimate tensile strength (UTS) and yield strength(YS) also decreased, while elongation increased. Tensile properties were sensitive to the build orientation, which indicates that DMLS generates anisotropy of material as a result of layered production and elongated β prior grains. It was noticed that inappropriate selection of parameters did not allow properties corresponding to the standards to be obtained due to the high porosity and defects of the microstructure caused by insufficient energy density.

## 1. Introduction

Modern implants must meet rigorous requirements for materials, machining technology and functionality. Implants, which replace the tissues of the human body, should have biomechanical properties comparable to those that are replaced and must not cause any side effects. The essential requirement for all medical implants includes corrosion resistance, biocompatibility, bio-adhesion, biofunctionality, machinability and availability [1,2,3,4,5]. To fulfill these requirements, materials being used for implants are tested for genotoxicity, carcinogenicity, reproductive toxicity, cytotoxicity, irritation, sensitivity and residues of sterilization [6,7]. Modern medical implants are regulated and classified in order to ensure the safety and effectiveness to the patient. Titanium and titanium-based alloys have been widely applied to medical materials, orthopedic implants and dental implants over the last few decades [8]. Among the different types of titanium alloys, Ti-6Al-4V remains the most widely used, as a material with a range of appropriate properties, such as higher strength, lower modulus of elasticity, better corrosion resistance and superior biocompatibility compared to other metallic biomaterials [9,10,11]. High corrosion resistance is primarily due to the spontaneous formation of the protective passive TiO_2_ film on titanium surfaces [12,13]. However, the durability of metallic implants in the body is limited. In the case of metal implants, a solid construction is associated with their high stiffness compared to bone tissue [14,15]. Disturbances in stress and strain distribution in the bone structure surrounding the implant lead to bone resorption around the implant and may lead to aseptic loosening of the implant. A good and lasting connection of the implant with the bone tissue is possible when there are sufficient conditions for the bone to grow into the pores of the material, therefore the use of a porous implant may be helpful in solving this problem [16,17,18].

Popularization of reconstruction procedures brings with it a number of new surgical challenges. Growing public awareness on the one hand, and the development of bioengineering sciences on the other, combine to create a significant increase in the demand and the availability of reconstructive procedures [19]. In response to the growing demand for custom-made implants, manufacturers are seeking to improve manufacturing processes by using advanced methods of preoperative planning. Modern design requires creating a physical, three-dimensional model of the designed element [20]. Conventional methods of preparing prototypes and models (e.g., casting, forging) are often extremely expensive and time consuming. New rapid manufacturing (RM) technologies, including direct metal laser sintering (DMLS) based on computer-assisted design, are applied to assist in this process [21]. Because of its ability to produce accurately and precisely objects, with different sizes, distribution of pores and complex geometry, this technology in the field of biomedical engineering is used increasingly [22]. An advantage of this process is the possibility to obtaining an arbitrarily shaped model by using laser beam. The laser sintering process is performed in the working chamber of a machine equipped with a computer that controls the production process [23]. Special software controls and regulates the pressure and atmosphere inside the chamber, depending on the material used. This process takes place by means of infrared radiation from a CO_2_ laser (10.6 μm) or Nd: YAG laser (1.06 μm) [24,25,26]. Laser scanning speed is defined as the speed at which the laser moves across the powder bed. The use of galvanic mirrors to reinforce the laser beam enables high scanning speeds by deflecting the beam [27]. Modern DMLS machines offer laser scanning speeds in the range of 0.1–10 m/s; however, in most studies on DMLS technology, speeds up to 0.1–1.5 m/s have been used up to now [24]. High-speed scanning conducive to building a faster and better process efficiency, but are associated with certain disadvantages [21].

Successful integration of an implant is generally accepted to rely on its surface characteristics such as chemical composition, morphology and energy [28]. Surface morphology is an important factor determining long-term implant stability, especially if bone quality is poor. A porous surface improves mechanical interlocking between the implant biomaterial and the surrounding natural tissue, providing greater mechanical stability at this critical interface [29,30,31,32]; however, DMLS allows only for control the macroscale porosity of produced parts, while micro- and nanoscales are not possible to control. Also, the DMLS method can induce non-metallurgical defects such as, e.g., pores of cracks. Surface roughness, pores—internal and external, uncontrolled residual stress and microstructure are drawbacks [33,34,35]. Accordingly, the associated mechanical properties and implementation of the produced element for a particular application may be inadequate. This paper defines the relationship between selected laser parameters, structure and mechanical properties of implant elements made from titanium alloy (Ti-6Al-4V, EOS GmbH, Munich, Germany).

## 2. Materials and Methods

Commercially available Ti-6Al-4V powder for sintering test samples, as the most widely titanium grade (Grade 5), was used. It is a two-phase α + β titanium alloy, with aluminum as the alpha stabilizer and vanadium as the beta stabilizer was used. β phase provides good mechanical properties, such as high strength and good ductility. α phase alloys have poor plasticity, but also have less tendency to absorb gases. The powder was produced by gas atomization. Spherical morphology and smooth surface indicated a good flowability and homogeneous layer distribution. The chemical composition of the powder, verified by Thermo ARL Quantris Spectrometer (Thermo Fisher Scientific, Waltham, MA, USA), diameter of particles, analyzed by ANALYSETTE 22 (FRITSCH, Idar-Oberstein, Germany) particle size analyzer and essential properties of the material, corresponding to ASTM F2924-14 and valid for powder material processed with EOSINT M 280/400 W (EOS Electro Optical System, Munich, Germany) are given in Table 1.

Dogbone tensile test samples were produced on the EOSINTM280 machine (EOS Electro Optical Systems, Munich, Germany), equipped with an Ytterbium fibre laser, using DMLS technology. For experimental work, six groups of test samples were performed. Each series was sintered with the same, constant laser power at 170 W, but with varying powder surface scanning speed. A scan speed between 300 and 1300 mm/s was chosen. 90° rotate scanning patterns with hatch distance of 0.1 mm for samples manufacturing was applied. The process parameters used in the present study have been illustrated in Table 2. The energy density (*E*) was calculated based on laser power (*P*), the distance between the laser scan line (*h*), scanning speed (*v*) and layer thickness (*t*) by means of equation [36].

(1)$$E=\frac{P}{vht}$$

Samples were sintered in a high vacuum environment, with the presence of argon and with oxygen content limited to 0.1%, which restricted the surface reactivity of the titanium compound. All samples were treated at 825 °C for 4 h in a vertical tube furnace, with a heating rate of approximately 10 °C/min, in an oxidation prevented argon atmosphere under furnace cooling.

Six series of tensile strength testing samples, of a size and shape conforming to standard PN-EN ISO 6892-1 (Figure 1) were performed. Each series were divided into two groups of samples—built in longitudinal (vertical) and in transversal (horizontal) configuration. The BrukerSkyScan 1172 (Bruker, Fitchburg, MA, USA) scanner was used for microtomographic studies of porosity with set parameters: number of rows—2664, number of columns—4000, Au + CU filter, sample rotation—0.2500, pixel size—4.28 μm. Scanned images of samples were reconstructed using programs dedicated to microCT image analysis—NRecon (1.2, Bruker, Fitchburg, MA, USA), Data Viewer (1.5.0.0, Ottawa, ON, Canada), CTvox (3.3, Bruker, Fitchburg, MA, USA) and CTAn (1.8, Bruker, Fitchburg, MA, USA). To remove all loose grains from the surface of the samples, they were rinsed in ethyl alcohol in an ultrasonic cleaner for 15 min. Tensile properties were studied using a Hegewald&Peschke INSPEKT (Meß- und Prüftechnik GmbH, Nossen, Germany) test machine with a maximum breaking strength of 5 kN. Displacements were measured by using an extensometer with a 25 mm gauge length. Yield stress and Young’s modulus were determined according to ASTM E 111. 

The preparation of samples for metallographic observations required grinding with abrasive paper with different grit gradation—from 200 to 2000 with two speeds—250 and 500 rpm, then polishing with 0.5 μm colloidal silica suspension and etching with freshly prepared Kroll’s reagent, consist of 5 mL of HNO_3_, 10 mL of HF and 85 mL H_2_O by immersion in the solution for 10 s, washing with distilled water and drying. Observations of the microstructure of the etched samples were carried out by using a Hitachi S-3000N (Hitachi, Tokyo, Japan) scanning electron microscope with an adapter for testing biological preparations. Vickers micro hardness measurements of the as-build and heat-treated, polished, unetched samples were performed, using Vickers hardness tester INNOVA analog stationary hardness tester (InnovaTest, Maastricht, The Netherlands). Hardness measurement was performed according to standard ISO 6507-1. For each sample five indentations were carried out at room temperature (20 ± 5 °C), under a load of 5 kgf (HV 5), in cross section longitudinal and transversal to the building layers were taken. Surface texture and roughness were investigated using an LEXT OLS4000 (Olympus, Tokyo, Japan) 3D measuring laser microscope. This equipment offers non-contact laser stylus tracing method. Surface topography scans were made on top (T) and lateral (S) surface of samples.

## 3. Results and Discussion

The top and lateral surfaces of the sample produced in the DMLS process are shown in Figure 2a,b, respectively. The frontal surface revealed the one-way pattern scanning. Measured distance between two adjacent, overlapping paths on which the laser beam moved was approx. 100 μm and corresponds to the sintering process assumptions and the proposed distances between the paths of the laser beam. These photographs also reveal the presence of pores and the loosely bound powder particles. The images depicting the side surfaces represent a completely different characteristic—strongly developed and almost entirely covered, only partially fused, grains of the powder, forming an approximately 100 μm thickness layer (Figure 2d). Non-melted or partially molten powder grains, loosely bound to a lower solidified layer (Figure 2b), make the surface of the sample very rough, which may be considered a porous structure. Non-melted grains have different diameters and their dimensional heterogeneity. The presence of so many unmelted or improper melted powder particles was caused by the fact that the powder used to produce the sample was not homogeneous. The surface roughness of the samples depends on the characteristics of the powder used—the morphology and size of the particles, but also on the power and speed parameters. Smaller powder particles are easier to melt (i.e., the laser power needed to melt them is smaller) compared to larger particles, due to the smaller surface area of the former. Different surface qualities were observed as a result of variable speed of the laser beam. For samples sintered at 300 mm/s (Figure 2c) and 500 mm/s (Figure 2d), the surface is smoother and free from defects visible on the surface of sintered samples at speeds of 1100 mm/s (Figure 2e) and 1300 mm/s (Figure 2f).

That characteristic surface certainly will affect the fatigue and corrosion resistance of the components. The roughness of a surface is considered to be a detriment to the fatigue properties of the component because it can serve as stress concentration and fatigue crack initiation site [37,38,39]. However, a rough surface may be beneficial for biomedical applications, such as facilitating the formation of bone structure of the surface [40]. The geometrically complex surface is a specific anchor for proteins and promotes cell adhesion, as well as regulates osteoblast differentiation and matrix production, thus accelerating the osteogenetic process. At the same time, it should be emphasized that in the case of DMLS technology, the porous structure of the surface is the result of the geometry of partially molten powder grains which, under the influence of interfaces between the surface of the implant and the tissue, can detach, resulting in acute and chronic effects. Moreover, such a high surface roughness will certainly affect the fatigue strength and corrosion resistance of sintered components, because surface irregularities will cause local concentration of stresses, which in consequence will initiate material cracking. Therefore, it is necessary to further modify the surface for homogenization.

Measured surfaces roughness described with Ra (surface average roughness), Rp (maximum peak height) and Rv (maximum valley depth) values are shown in Table 3. The roughness of the lateral and upper surfaces was compared with the energy density to analyze the relationship between them. The summary shows that the increase in energy density in the range from 88 to 113 J/mm^3^ reduces the roughness of both surfaces, while the energy density in the range from 63 to 44 J/mm^3^ influences its increase. The increase in surface roughness of samples at the highest energy densities results from the intensification of particle agglomeration processes on the surfaces of the analyzed samples, which is confirmed by the photographs taken using a scanning microscope (Figure 2).

Scanning speed in the range of 500–900 mm/s leads to the generation of stable integration paths. The shape of individual paths is quite clear, their surface is even, and the width is constant. The surface of the samples is smooth, and the number of pores resulting from the local overheating and evaporation of the material is smaller. An increase in the scanning speed to 1100 mm/s reduces the energy density (52 J/mm^3^). Under these conditions, it is difficult to achieve a melting point that allows the powder particles to melt completely, while the amount of liquid phase is insufficient to form a continuous, stable layer. The sample surfaces are clearly rough with visible discontinuities. The unmelted particles on the surface are additionally deformed by a moving reversible blade, and some of them are ripped out and transferred, which results in an increase in porosity. For samples sintered at 1300 mm/s, the effect of balling is visible, induced by scanning speeds that are too high, and a short time of interaction of the laser beam with the powder. With such a high speed it is impossible to maintain the path continuity. The effect of spinning is caused by the lack of wetting of the molten pool with the previous layer, which affects cohesion adversely, thus making the subsequent layers more difficult to bond.

The occurrence of the spinning effect causes deterioration of the surface resulting from a combination of thermal stresses and poor inter-layer bonding between the powder particles and subsequent layers [41]. Despite the combination of particles, the mechanical strength of components from this group is very low, because the samples produced using the proposed technological parameters have many disadvantages, and they are characterized by high porosity and low fracture toughness, which was confirmed later in the work. 

Due to the shear forces present at high scanning speeds, high surface tension may occur within the pool. Melted surfaces, to reduce surface tension, are subject to the process of balling. This phenomenon occurs when the molten material does not wet the underlying substrate because of its surface tension, which tends to spheroidize the liquid [39]. This means that the density of energy does not always increase the relative density of the sample and there is a limit above which the results deteriorate. On the other hand, energy density values that are too low are not suitable for ensuring adhesion between successive layers, because the depth of penetration is not sufficient [40]. Largegrains, unmelted but entrapped in the solidifying material, which protrude above the sample surface, increase the surface roughness. An important issue at high scanning speeds is also that high shear forces can eject pieces of molten metal from the melt pool, thereby causing the formation of internal pores and a reduction in the final density of the product [41,42,43].

For samples sintered with 170 W laser power at low scanning speeds from 300 to 700 mm/s, the porosity of the material changes from 2.59% down to 1.39%. Above 900 mm/s an increase in porosity is observed from 9.29% at the speed of 900 mm/s to 12.41% at the speed of 1300 mm/s. Sintered samples with a power of 170 W achieved a density of over 95% for three speeds (500, 700, 900 mm/s). The highest density—99.23%—was obtained in samples scanned with a 170 W laser at 500 mm/s (Figure 3).The research shows that the increase in porosity is accompanied by a decrease in energy density. Differentiated porosity morphology, conditioned by variable parameters—laser power and time of exposure of the powder material to the laser beam—was revealed in optical studies. Detailed analysis of porosity is shown in Figure 4, where the relationship between energy density (J/mm^3^) and porosity (%) was indicated. At low energy density of the laser (44 J/mm^3^–63 J/mm^3^), the porosity ranges from 12.49% to 9.29% The pores are unevenly distributed, irregular in shape and interconnected. Porosity is characterized by large voids filled with loose particles of unmelted grains—lack of fusion. A possible explanation for this is that with such a low energy density of the laser and due to the smaller depth of laser penetration, the size of the melt pool is too small and the powder particles are not sufficiently fluidized to provide sufficient bond between the layers. Increased laser energy density in the range from 81 J/mm^3^ to 113 J/mm^3^ reduces porosity, which varies from 1.39% to 0.77%. Increasing the energy density of the laser generates a relatively high temperature, which facilitates the flow of liquids and filling the pores (voids). This is due to the relatively low viscosity of the alloy, and thus the density of the material increases after solidification. At higher energy densities, the pores are small and mostly spherical, and their presence is the result of solidification shrinkage and gas bubbles trapped by the molten powder. An increase in laser energy density above 188 J/mm^3^ increases the porosity to 2.59% and changes its morphology. High energy density of the laser can cause the evaporation of small grains with low surface energy, leaving them empty, as pores, within the built layer or between layers. Any impurities such as oxides present on the surface of the powder particles are also a source of gas porosity and local delamination [36,40,43].

Li et al. [44] confirmed the relationship between roughness and energy density. Complete sintering of the particles may be achieved with a lower speed laser that induces higher local temperature, lower viscosity and contact angle, higher Marangoni flow and capillary forces. This, however, effects on full densification and lower surface roughness. Higher scanning speeds, on the other hand, generate lower energy density and only partial melting of the powder. The viscosity of the molten material is quite high, which leads to the balling effect and higher roughness.

The porous structure described was also discussed by [39]. They analyzed various variants and parameters of the sintering process of stainless steel. In the course of the research, they found that properly selected laser power and scanning speed ensure continuity of scanning paths and permanent inter-layer bonds, which results in obtaining full density structures. The continuation of their research proved that scanning speeds that are too high considered at constant laser power cause discontinuities in the scanning paths and the effect of balling. Lower sintering powers at higher scanning speeds generated a porous structure with an open morphology, with a significant amount of the pores being formed on the surface of the samples.

The DMLS process binds the material in layers, so if the output energy is not adjusted to the properties of the powder and other relevant process parameters the pool width is too small for the scanning paths to overlap. This is the main reason for the non-melted powder particles remaining in the sample. This phenomenon is called lack-of-fusion. Usually, its occurrence is located between the two scan tracks and the deposited layers. The presence of cumulative inter-layer defects and stress resulting from the sintering process may result in the spread of defects, thus affecting the deterioration of the final properties of the product.

The microstructure of titanium alloy samples manufactured by using DMLS technology differs from the microstructure of wrought alloy due to different cooling rates. Ti-6Al-4V obtained by DMLS technology has a fine-grained martensitic structure (α’). This may be attributable to high-temperature heating and cooling, which are inherent in the DMLS process. Repeated heating and cooling cycles result in an increase in diverse morphologies, leading to visible band structure (Figure 5c, indicated by white arrows) [45]. A characteristic feature of laser sintered materials are columnar prior grains β, disclosed on the XZ plane across several deposited layers, arranged parallel to the construction direction (Figure 5a,b). These columnar grains arise from the epitaxial growth of the prior β grains in previous layers due to the high-temperature gradient and solidification rate in the molten pool. The presence of prior β columnar grain boundaries is due to the fact that Ti-6Al-4V solidifies in the β phase field and heat is mainly conducted away vertically. The average width of the prior β grains was 101 ± 14 μm, which corresponds to the optimised hatch spacing (100 μm). Optical microscopy analysis reveals that the α′ grains did not vary in size along the build and scanning directions. After heat treatment these grains, although less clearly, were still visible.

The microstructure parallel with deposited layers before and after annealing was equiaxed polyhedric with visible porosity (Figure 5c,d). Also, the chessboard pattern was identified on the XY plane, which is the effect of the scanning strategy (Figure 2d). Many studies have shown that the α‘ microstructure causes low ductility, determined with high tensile strength [13,27,28]. After heat treatment the fine metastable martensitic structure has been transformed into a mixture of a fully lamellar α and β and leads to structure pattern dissolving. 

Decomposition of martensite resulted in changes in strength and plasticity. The size of α grains was coarser (α lath width = 1.55 ± 0.223 μm) in this case as compared to the “as built” Ti-6Al-4V (α’ lath width = 0.593 ± 0.261 μm). As a result of martensite decomposition, the alloying elements are redistributed. The β phase is enriched with vanadium and phase α-aluminum. 

The elongation, tensile strength and Young’s modulus for as-built and heat treated alloy samples are given respectively in Figure 6 and Figure 7. Presented results show that the change in energy density resulting from the change in laser speed had a significant impact on a final tensile strength, yield strength or Young’s modulus. Those values are significantly higher than the required values for cast and forged material due to martensitic microstructure, as well as Young’s modulus (according to ASTM F1108-14 and ASTM F136-13). However, the elongation of as-built samples with an average value of 2.1–5.5% is much lower than required. Poor ductility is a result of martensitic microstructure. After heat treatment at 850 °C, the ultimate tensile strength (UTS) and the yield strength (YS) of the alloy decreased(UTS from ~1158 MPa to ~924 MPa, YS form ~1122 MPa to ~805 MPa), but the ductility is greatly increased form 4–5.5% to 12.6–16.0%. This is due to changes in the microstructure and decomposition of brittle martensite, which is transformed at a temperature of more than 800 °C into a more plastic α and lath size increase after a relatively slow cooling rate [46]. For both as-built and heat treated samples, the highest strength among them is characterized by samples sintered with 500 mm/s, which correlates with the lowest porosity. The lowest tensile strength was demonstrated by samples with the highest porosity, sintered at 1300 mm/s. It should be noted, that after heat treatment UTS and YS are similar to wrought and annealed Ti-6Al-4V (930 MPa UTS, 860 MPa YS) for samples built with scanning speed 300 and 700 mm/s, also samples built with scanning speed 900 and 1100 mm/s meets standards for casted material (860 MPa UTS, 760 MPa YS).

Both materials—as-built and heat-treated—are characterized by anisotropy; samples built in Z (vertical) direction have higher UTS, YS and elongation at break compared to the X (horizontal) direction (Figure 7). The correlation may be explained by the microstructure of the material and the direction of growth of epitaxial β-grains. The columnar boundaries of these grains in the samples built in the vertical direction are arranged parallel to the tensile direction, while in the samples built in the XY plane these boundaries are perpendicular. This is also an explanation for the lower elongation obtained in the horizontally built samples.

Anisotropy of elastic modulus as a result of crystal anisotropy was reported by References [26,47,48]. Also, the Young’s modulus did change with the building direction and orientation of the prior β-grains. It should be emphasized that for the as-built samples, the Young’s modulus was higher in the Z direction by approx. 15% in relation to the X direction, while after the heat treatment the difference between the Young’s modulus values was insignificant. The same applies to both UTS and YS changes. The decrease in the Young’s, UTS and YS modulus as well as the increase in elongation after heat treatment indicate an increase in the plasticity of the material, which is also confirmed by changes in the microstructure.

To assess the anisotropy of mechanical properties, the following equation [49] was used to calculate the differences of properties between horizontally and vertically DMLS-built samples:(2)$$Dr=\;\frac{\left|P_{V}-P_{H}\right|}{\left|P_{V}+\;P_{H}\right|/2}\;\times \;100\%$$
where *Dr* is a difference ratio of analyzed properties, $P_{V}$ and $P_{H}$ represent horizontal and vertical property of material. Anisotropy of tensile strength, yield strength, elongation and elasticity modulus is summarized in Table 4.

Anisotropic properties in laser fused materials result from the presence of column structures whose boundaries affect the tensile strength and elasticity of the material. Untreated samples are characterized by a distinct anisotropy of the modulus of elasticity and elongation. As expected, heat treatments effectively alleviate the anisotropy of mechanical properties as an effect of microstructural changes.

The study of the fractography of fracture surfaces of impact strength specimens showed some imperfections of the DMLS process (Figure 8). Voids and un-melted powder (attributed to entrapped gas in the melt pool and lack of melting during fabrication) can be seen along the fracture surface of specimens in both as-built (Figure 8a,b) and heat-treated conditions (Figure 8c,d), as well as horizontal and vertical samples. Fracture surface of non-treated samples showed a brittle character, accompanied by very small plastic deformation and energy absorption. Untreated DMLS Ti-6Al-4V samples indicate shallow dimples, cleavage facets, ledges and terraces that suggest cleavage most probably along the brittle α’ needles characteristic for brittle fractures. Fracture surface of horizontal samples indicate a large amount of defects and gas pores, which caused stress concentration for cracks growth. Characteristics of the breakthrough after heat treatment shows a mixed mode brittle and ductile character. Ductile cracking is the effect of martensite decomposition and appearance in the microstructure ductile and plastic β phase. It can be seen that the morphology of the tensile fracture resembles honeycombs with a number of small voids. There are no significant differences between the fracture surfaces of differently oriented samples.

Hardness values for as-build and heat-treated Ti6Al4V samples are shown in the graph in Figure 9. The hardness of sintered materials depending on the laser power measured in two directions—longitudinal and transversal. The measured results showed that the decrease of laser speed and energy density led to decrease in hardness of materials in both directions. The influence of heat treatment on hardness is clearly visible. The as-built material is harder than the treated one. This phenomenon was expected due to relationship between hardness and microstructure. As-built material has a α’ martensitic microstructure—much harder than the laminar α + β microstructure received after heat treatment.

## 4. Conclusions

This work focuses attention on the influence of input process parameters on the mechanical parameters of Ti-6Al-4V alloy powder, which is highly biocompatible. Samples were manufactured in a controlled atmosphere with inert gas and therefore had no contact with any contaminants, as is the case in traditional methods such as turning and milling, but even so—the DMLS component contains flaws such as internal and external pores, unmelted or semi-melted powder particles, inclusions and a surface finish with relatively high roughness. The surface area of the samples obtained during the DMLS process is heterogeneous, rough and porous. This is due to imperfections in the process, difficulties in choosing the right parameters, and the quality of the powder. Too low a speed of the laser beam has led to an increase in the power density and evaporation of small powder particles, which solidify falling on the molten layer of material, generating roughness. However, too high a speed caused the energy density to be insufficient to properly melt the grains, which caused unstable and discontinuous paths. The best surface quality, characterized by the smallest roughness of both surfaces, were sintered samples with laser speeds of 500 and 700 mm /s. The lowest porosity was also achieved for scanning speeds of 500 and 700 mm/s. The generated energy density was sufficient in this respect to melt the material properly. The porosity obtained for these two speeds did not exceed 1%. The energy density above 160 J/mm^3^ and below 75 J/mm^3^ affected a significant increase in porosity in the material obtained, with the morphology of the resulting pores being clearly different.

There are distinct differences in the microstructures of samples manufactured by DMLS and wrought material. The microstructure obtained as a result of rapid solidification after selective laser sintering reveals a very fine morphology with α particles within β boundaries. The presence of martensite or α’ is also evidence that the material has undergone rapid cooling during the laser sintering process. The influence of high temperature and annealing time caused a change of martensitic microstructure to two-phase α + β with different mechanical and plastic properties.

Both porosity and microstructure had an effect on mechanical properties. Because of the very fine microstructure, the yield stress and ultimate tensile stress of the as-built samples were quite high. The change of microstructure and decomposition of martensite under the influence of heat treatment caused the strength properties to be reduced, while the plastic properties increased. The tensile properties of the samples characterized by the lowest porosity content, after heat treatment, satisfied the requirements of standard ISO 5832-3. The anisotropic properties of the material were more pronounced for raw samples, while after heat treatment the properties were uniform. Higher tensile strength was characterized by vertical samples. Theoretically, due to the layered structure, this orientation should weaken the strength; however, the presence of epitaxial grain boundaries caused an increase in the tensile strength in the direction of the sample building.

The results of the hardness test of the samples also indicate the anisotropic properties of the material. Furthermore, as the scanning speed decreases, hardness decreases in both directions. After heat treatment—as a result of changes in the microstructure—a decrease in hardness is noticeable; however, it is much larger for samples built in the vertical direction. The decrease in hardness is insignificant for samples built horizontally.

Analysis of the results obtained allows for the conclusion that modifying the technological parameters of the sintering process makes it possible to change the properties of the element obtained, depending on the requirements of the product. Appropriate selection of settings for the laser and atmosphere in the working chamber is the key factor in achievement of satisfactory results in applications of DMLS technology. Future research should be focused on the mechanical properties of DMLS samples and fatigue testing, porosity and defects, the effects of heat treatments, and corrosion resistance.

## Figures and Tables

**Figure 1 materials-12-00176-f001:**
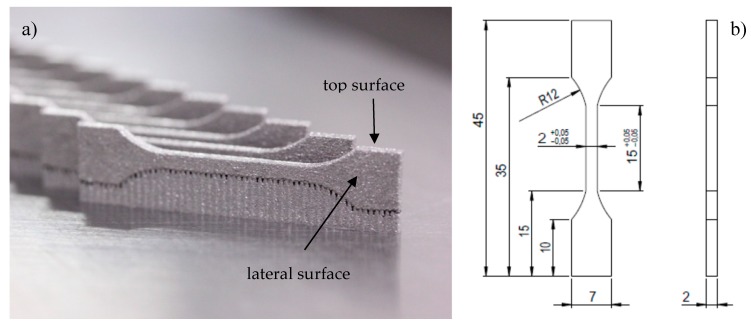
Samples: for tensile test (**a**); dimensions in mm (**b**) of tensile samples.

**Figure 2 materials-12-00176-f002:**
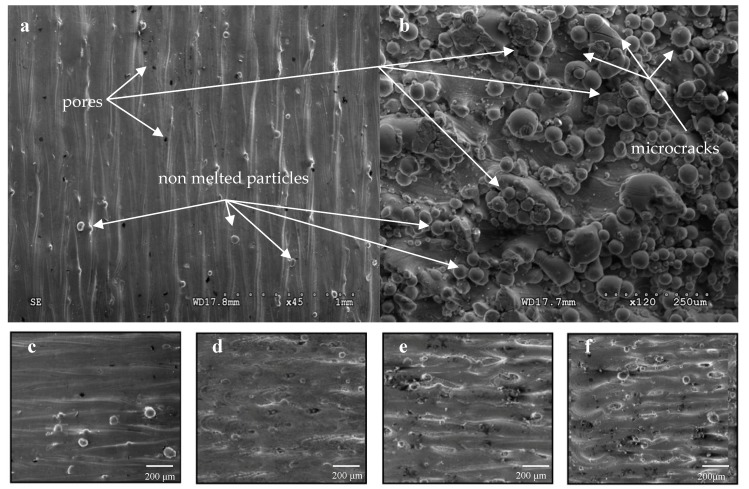
Top surface of a vertically built direction (**a**), lateral surface (**b**) and top surface of sample sintered with laser beam speed 300 mm/s (**c**), 500 mm/s (**d**), 1100 mm/s (**e**), 1300 mm/s (**f**).

**Figure 3 materials-12-00176-f003:**
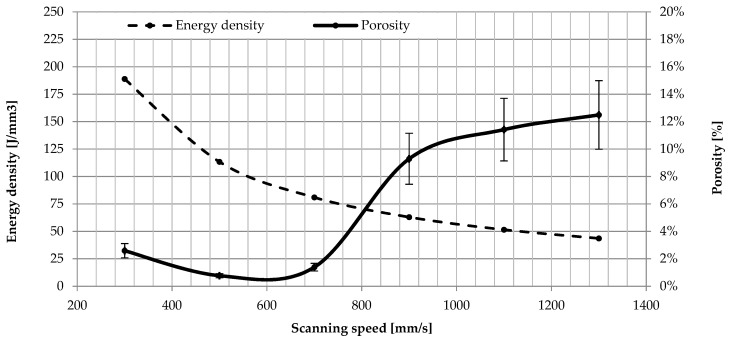
The influence of scanning speed and energy density on the porosity.

**Figure 4 materials-12-00176-f004:**
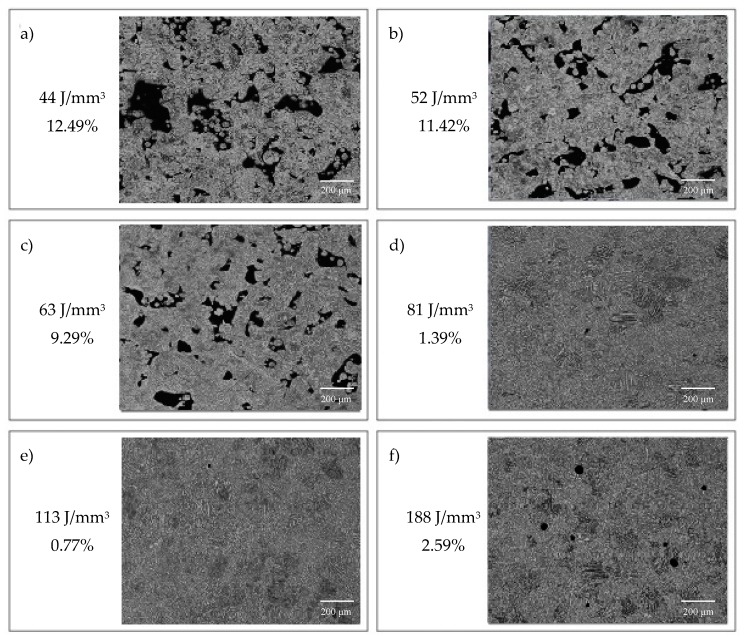
The influence of energy density on the porosity and their morphology: 44 J/mm^3^ (**a**), 52 J/mm^3^ (**b**), 63 J/mm^3^ (**c**), 81 J/mm^3^ (**d**), 113 J/mm^3^ (**e**), 188 J/mm^3^ (**f**), mag. 100×.

**Figure 5 materials-12-00176-f005:**
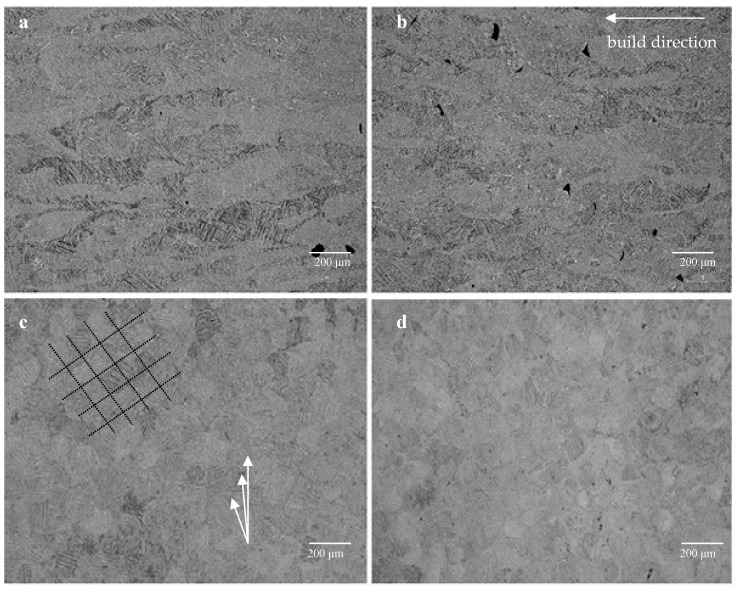
Cross-sections perpendicular (**a**,**b**) and parallel with deposited layers (**c**,**d**) before (**a**,**c**) and after (**b**,**d**) heat treatment, revealing microstructure of the as built (**a**,**c**) and annealed sample (**b**,**d**); mag. 100×.

**Figure 6 materials-12-00176-f006:**
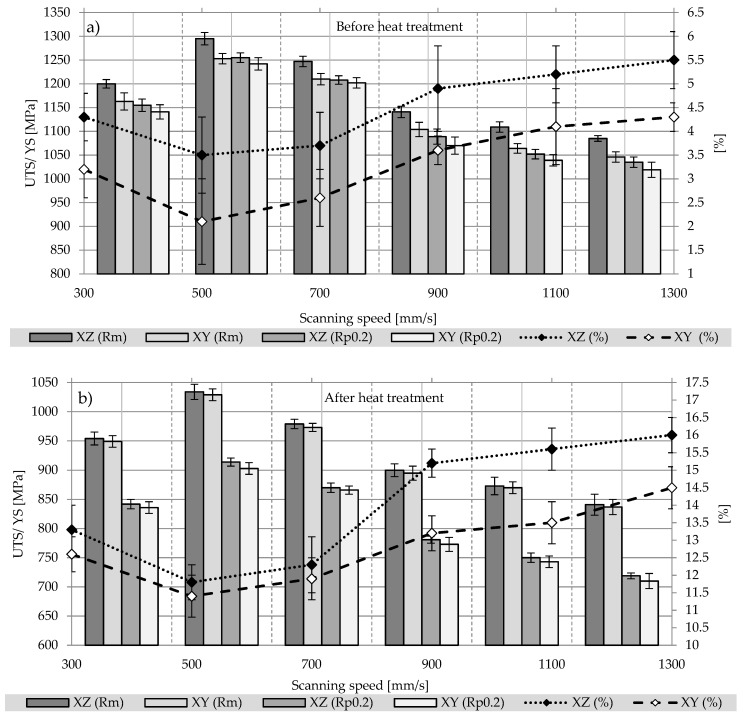
Mechanical properties of Ti6Al4V samples vertically (XZ) and horizontally (XY) oriented to the building direction before (**a**) and after (**b**) heat treatment.

**Figure 7 materials-12-00176-f007:**
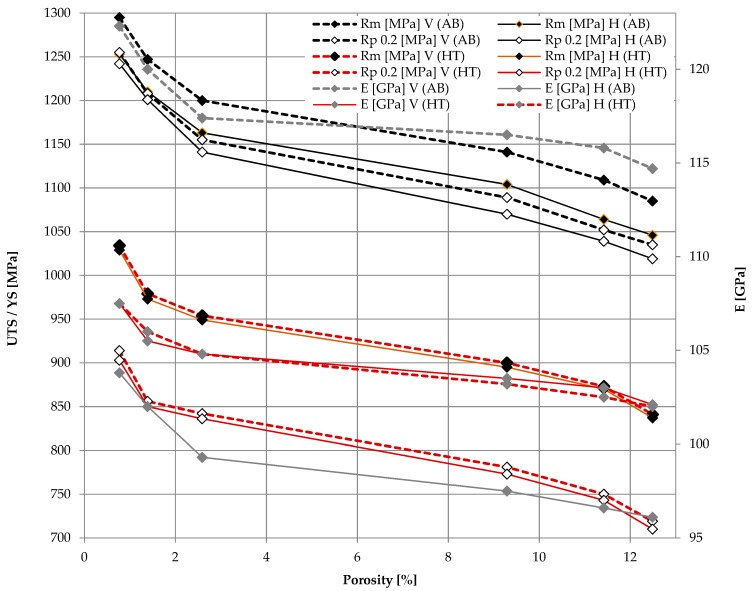
Anisotropic properties of samples built with different orientation.

**Figure 8 materials-12-00176-f008:**
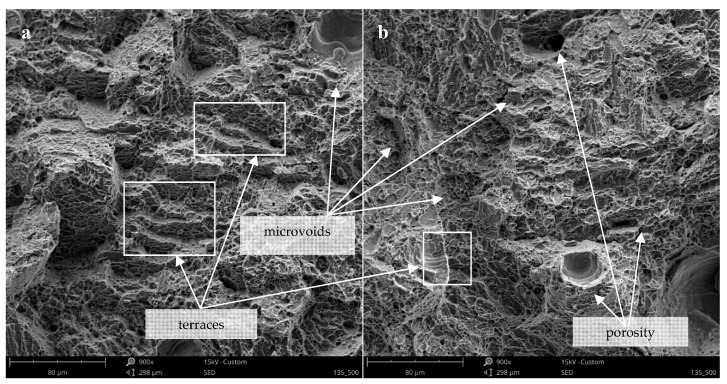
Fracture surface of non-treated (**a**,**b**) and heat treated (**c**–**f**) vertically (**a**,**c**,**e**) and horizontally (**b**,**d**,**f**) oriented samples; mag. 900× and 1500×.

**Figure 9 materials-12-00176-f009:**
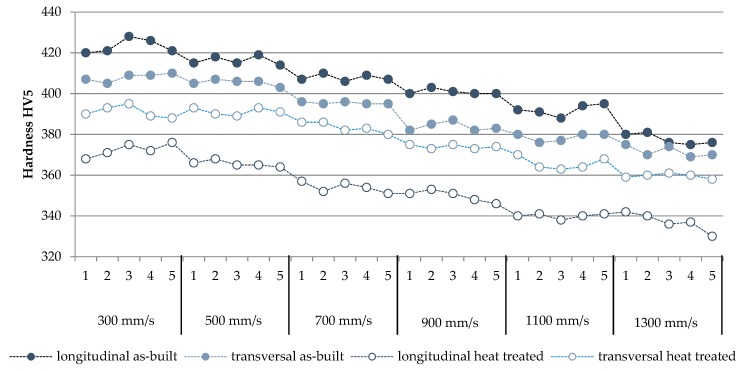
Comparison of series and samples hardness.

**Table 1 materials-12-00176-t001:** Chemical composition and material characteristics of used powder.

**Element**	Al	V	O	N	H	Fe	C	Ti
**Composition (wt %)**	5.97	4.04	0.195	0.036	0.010	0.24	0.061	Bal.
**ASTM F136**	5.5–6.75	3.5–4.5	<0.2	<0.05	<0.015	<0.3	<0.08	Bal.
**Density**	4.41 kg/dm^3^							
**Melting Point**	1670 °C							
**Diameter of Particles**	10–400 μm							
**Powder Layer Thickness**	30 μm							
**Hatch Distance**	100 μm							

**Table 2 materials-12-00176-t002:** Processing parameters.

**Laser Power (W)**	170
**Scanning Velocity (mm/s)**	300
	500
	700
	900
	1100
	1300
**Energy Density (J/** **mm^3^)**	189
	113
	81
	63
	52
	44
**Layer Thickness (mm)**	0.03
**Spot Size (mm)**	0.1
**Hatch Distance (mm)**	0.1

**Table 3 materials-12-00176-t003:** Roughness parameters.

Scanning Speed [mm/s]	Ra [µm] (Surface Average Roughness)				Rp [µm] (Maximum Peak Height)				Rv [µm] (Maximum Valley Depth)			
	AB		HT		AB		HT		AB		HT	
	T	S	T	S	T	S	T	S	T	S	T	S
300	5.31	11.78	5.01	10.94	12.86	26.08	12.35	25.61	11.32	16.96	11.25	16.77
500	3.76	9.19	3.48	8.32	10.23	20.72	10.02	20.44	10.11	18.01	10.08	17.90
700	4.06	10.35	3.92	9.45	10.56	21.31	10.17	20.88	10.07	20.68	9.92	19.82
900	5.49	17.25	5.05	16.15	13.17	27.01	12.89	26.58	12.81	26.21	12.41	26.11
1100	6.55	20.06	6.23	19.15	15.32	29.70	15.15	29.33	13.36	31.41	31.27	31.26
1300	10.22	21.06	9.91	20.62	19.21	38.01	19.03	36.29	15.41	35.51	15.30	35.33

AB as built, HT heat treated, T top surface, S lateral (side) surface.

**Table 4 materials-12-00176-t004:** Anisotropy of tensile strength, yield strength, elongation and elasticity modulus before and after heat treatment.

Material	*Dr* UTS %	*Dr* YS %	*Dr* EM %	*Dr* El %
As-built	3.36%	0.22%	17.11%	29.55%
HT	0.5%	0.97%	0.08%	8.80%

*Dr*—difference ration of analyzed properties; UTS—Ultimate Tensile Strength; YS—Yield Strength; EM—Elastic Modulus; El—Elongation.

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
