# Peer review of "Mechanical Properties and Microstructure of DMLS Ti6Al4V Alloy Dedicated to Biomedical Applications"

_materials, 2019, doi:10.3390/ma12010176_

Reviewer 1 Report

This paper is new, original and well organized.  English language is good in the paper and all references are adequate. Also all parts of paper are important and conclusion is fine. I recommend this paper for publication.

Author Response

Sir/Madame

The authors of the work would like to thank for the positive review of this article and recommendation this paper for publication.

Zaneta Anna Mierzejewska

Radovan Hudak

Jaroslaw Sidun

Reviewer 2 Report

This manuscript is well organized, well written and scientifically sound. The authors did a great job introducing the research subject and discussing the acquired data. My only suggestion is to reduce the conclusions sections since there is a lot of repetition of information; they should be presented in a more concise manner.

Author Response

Sir/Madame

The authors would like to thank for the positive opinion and the remark concerning the extended the conclusions sections of the article.

Conclusion section contains the most important information that was included in the research part of the article. It presents the results and short comments that can actually give the impression of repetition in the content of the article, but according to the authors, presenting them in this way highlights the essence of the research. This section contains the main conclusions and consequences of them in relation to the research problem presented in the article. We believe that the content provided in this section is interdisciplinary, understandable for a wider circle of scientists and interested persons without the need to analyze the whole article. The authors are open to further suggestions and if the reviewer still considers it necessary to analyze and rebuild this part, the authors will attempt to implement the reviewer's comments.

Reviewer 3 Report

The paper entitled "Mechanical properties and microstructure of DMLS Ti6Al4V alloy dedicated to biomedical applications” Anna et al. deals with the study of the production of Ti6Al4V for biomedical applications using DMLS. The microstructure or the parts, quality (porosity, roughness), and mechanical properties (strength, hardness and fracture surfaces) were analysed.  

The results are in the scope of Materials journal. These are interesting; however, after reading the paper, I have some comments about it:

GENERAL COMMENTS:

1)     This paper suggest that a DMLS strategy has been applied however, it seems that parts were produced by SLM. Powder was fully melted rather than sintered. Therefore, authors should clarify which approach was followed. In fact, DMLS and SLM have been used along the manuscript.

2)     Parts produced in this way are intended to be used in biomedical applications; however, no in-vitro or in-vivo experiments were included. The potential of this application should be demonstrated; and the discussion of the results should take into account this application.

PARTICULAR COMMENTS

1)     (Line 34) Please, replace “machanability” with “machinability”

2)     (Lines 72-76) It is true that surface properties are relevant for the implant osseointegration (especially, topography and wettability, see e.g.: Riveiro, Antonio, et al. "Laser surface texturing of polymers for biomedical applications." Frontiers in Physics 6 (2018): 16.); however, it should be noticed that different scales are involved. Micro- and nanoscales are as relevant as the macroscale. DMLS can control the macroscale in produced parts; however, the micro- and nanoscales are not possible to be controlled during DMLS processing.

3)     (Line 77) DMLS can induce non-metallurgical defects such as, e.g.: pores, or cracks.

4)     (Line 111) Please, include the focal length, focus spot position. How was measured the spot diameter?

5)     (Materials and methods) Please, include details about the cleaning procedures for the workpieces after DMLS.

6)     (Materials and methods) Please, include more details about the fabrication strategy. There are no indications about how the sample was fabricated: vertically, horizontally: however, in the results section it is suggested that different part orientations were studied.

7)     (Lines 147-148) Non melted particles can produce a detrimental effect for biomedical applications. These particles can be released during the lifetime of the implant and can produce inflammations or the implant failure (see, e.g.: Hallab, N. J., & Jacobs, J. J. (2009). Biologic effects of implant debris. Bulletin of the NYU hospital for joint diseases, 67(2), 182.)

8)     (Line 148) It is asserted that the presence of these non-melted particles on the surface resembles a porous structure. Bone growth is not as simple here as in presence of cavities because these particles can be released.

9)     (Line 190) Please, include the meaning of AB, HT, T and S in the Table caption

10) (Lines 209-234) How was the porosity measured? Only from image analysis or micro-CT measurements were performed?

11) (Line 234) Please add error bars to Fig. 3.

12) (Line 234) Please, change “Porosity [[%]” with “Porosity [%]” in the y-axis of Fig. 3.

13) (Lines 261-275) Was the microstructure analysed along the whole parts? As pointed out in the paper, the heat extraction is vertical; furthermore, there is a heat accumulation effect over passes; therefore, the solidification time and thermal gradients can change along the workpiece, and in consequence the microstructure.

14) (Line 301-302) Please, include the titles of the y-axis for Fig. 6.

15) (Line 321) Please, include error bars in Fig. 7

16) (Line 333) SLM or DLMS? Please, clarify which approach was followed.

17) (Line 375) Please, include error bars in Fig. 9

Author Response

Response to Reviewer 1 Comments

GENERAL COMMENTS:

Point 1: This paper suggest that a DMLS strategy has been applied however, it seems that parts were produced by SLM. Powder was fully melted rather than sintered. Therefore, authors should clarify which approach was followed. In fact, DMLS and SLM have been used along the manuscript.

Response 1:

There are various acronyms and names used for the process of using a direct metal laser sintering 3D printer, and

these can be somewhat confusing.In fact, there is some debate even amongst the experts as to what terms should

 be used. Many literature sources state that SLM and DMLS processes differ due to the particle binding mechanism.

On the other hand it is common  statement in scientific articles that this particular metal-printing processes is known

as SLM or DMLS.

The process is known by both names, depending on the geographical area of the user - SLM was most commonly

used in Europe and DMLS in the USA, but both names have been used synonymously. In fact, DMLS, SLM as well

as Laser Cusing are proprietary trade names of the same technology of selective remelting of powdered metals with

a laser, applied layer by layer, until a ready-made fully durable part is obtained. Initially, the name was used in parallel

by MCP Hek and EOS. However, the first company reserved it, and the German EOS was forced to invent its own

name - DMLS. Concept Laser, on the other hand, defines this technology as Laser Cusing.

Since the samples printed for the purpose of this work and tests were made using the EOS INT machine, the name of

the process reserved for this company was used.

Point 2: Parts produced in this way are intended to be used in biomedical applications; however, no in-vitro or in-vivo experiments were included. The potential of this application should be demonstrated; and the discussion of the results should take into account this application.

Response 2:

Indeed - the title of this article indicates that the studies in it relate to biomedical applications. The intention of the authors was to focus on the important strength parameters of the material used in hard tissue replacements, i.e. Ti6Al4V titanium alloy, which is considered biocompatible - hence biomedical applications in the title. On the basis of the results presented in this work and the optimal parameters of consolidation of the powder particles, another article is created in which both in-vivo and in-vitro test results will be presented. The tests carried out for this purpose include: corrosion tests in selected solutions simulating biological fluids, surface analysis and crystals deposited on it, as well as tests on living human cells, in order to assess their viability and surface overgrowth.

PARTICULAR COMMENTS:

Point 1: (Line 34) Please, replace “machanability” with “machinability”

Response 1: The mistake has been corrected in the manuscript.

Point 2: (Lines 72-76) It is true that surface properties are relevant for the implant osseointegration (especially, topography and wettability, see e.g.: Riveiro, Antonio, et al. "Laser surface texturing of polymers for biomedical applications." Frontiers in Physics 6 (2018): 16.); however, it should be noticed that different scales are involved. Micro- and nanoscales are as relevant as the macroscale. DMLS can control the macroscale in produced parts; however, the micro- and nanoscales are not possible to be controlled during DMLS processing.

Response 2: Reviewer is right, and his remark about the surface properties has been included in the article. The analyzed articles and research show that it is possible to control the porosity of the material on a macro scale, which allows the creation of scaffolds for tissues, lighter production of openwork elements. However, it is impossible or very difficult to control the internal porosity, the level of which can only be minimized by using a powder with appropriate granularity and homogeneity, free of impurities and selection of appropriate melting parameters.

Point 3: (Line 77) DMLS can induce non-metallurgical defects such as, e.g.: pores, or cracks.

Response 3: The mistake has been corrected in the manuscript.

Point 4: (Line 111) Please, include the focal length, focus spot position. How was measured the spot diameter?

Response 4: In the used device EOS INT F-theta and the entirety of the optical system is automatic and defined by the manufacturer during the device calibration. Focal length is unchanged and reaches 410 mm. Diameter of laser beam at building area is variable in the range from 70 to 120 µm, however in calibration process the spot diameter was set to 100 µm and remained unchanged.

Point 5: (Materials and methods) Please, include details about the cleaning procedures for the workpieces after DMLS.

Response 5: The information the reviewer requested was completed in the manuscript.

Point 6: (Materials and methods) Please, include more details about the fabrication strategy. There are no indications about how the sample was fabricated: vertically, horizontally: however, in the results section it is suggested that different part orientations were studied.

Response 6: The orientation in which the samples for research was built was presented in part 2. Materials and Methods (line 116), according to the reviewer's note was supplemented and highlighted.

Point 7: (Lines 147-148) Non melted particles can produce a detrimental effect for biomedical applications. These particles can be released during the lifetime of the implant and can produce inflammations or the implant failure (see, e.g.: Hallab, N. J., & Jacobs, J. J. (2009). Biologic effects of implant debris. Bulletin of the NYU hospital for joint diseases, 67(2), 182.)

Point 8: (Line 148) It is asserted that the presence of these non-melted particles on the surface resembles a porous structure. Bone growth is not as simple here as in presence of cavities because these particles can be released.

Response 7 and Response 8:

The reviewer's comment on this issue is very right. Indeed, the information contained in the literature section was not presented correctly, raising some doubts and a fair discussion.

The article has been supplemented with the following commentary:

The geometrically complex surface is a specific anchor for proteins and promotes cell adhesion, as well as regulates osteoblast differentiation and matrix production, thus accelerating the osteogenetic process. However, at the same time it should be emphasized that in the case of DMLS technology, the porous structure of the surface is the result of the geometry of partially molten powder grains which, under the influence of interfaces between the surface of the implant and the tissue, can detach, resulting in acute and chronic effects. Moreover, such a high surface roughness will certainly affect the fatigue strength and corrosion resistance of sintered components, because surface irregularities will cause local concentration of stresses, which in consequence will initiate material cracking. Therefore, it is necessary to further modify the surface for homogenization.

Point 9: (Line 190) Please, include the meaning of AB, HT, T and S in the Table caption

Response 9: The mistake has been corrected in the manuscript

Point 10: (Lines 209-234) How was the porosity measured? Only from image analysis or micro-CT measurements were performed?

Response 10: Porosity studies were carried out using a Bruker SkyScan 1172 microtomograph. The entire sample was scanned, then the images obtained were reconstructed with using programs dedicated to microCT image analysis, which allowed for automatic calculation of porosity in the entire sample volume.

Point 11: (Line 234) Please add error bars to Fig. 3.

Response 11: The mistake has been corrected in the manuscript

Point 12: (Line 234) Please, change “Porosity [[%]” with “Porosity [%]” in the y-axis of Fig. 3.

Response 12: The mistake has been corrected in the manuscript

Point 13: (Lines 261-275) Was the microstructure analysed along the whole parts? As pointed out in the paper, the heat extraction is vertical; furthermore, there is a heat accumulation effect over passes; therefore, the solidification time and thermal gradients can change along the workpiece, and in consequence the microstructure.

Response 13: The microstructure study was carried out in two places - 2 mm and 20 mm above the breakthrough. The surfaces were prepared in accordance with the information contained (and supplemented) in point 2. Materials and Methods. Differences in microstructure were unnoticeable. However, the authors are aware of the importance of the factors indicated by the reviewer and will strive to deepen the knowledge about changes in the microstructure analyzed using more modern microscopes, which at the time of writing the article was not possible due to limitations in terms of equipment.

Point 14: (Line 301-302) Please, include the titles of the y-axis for Fig. 6.

Response 14: The mistake has been corrected in the manuscript

Point 15: (Line 321) Please, include error bars in Fig. 7

Response 15: Fig. 7 showing the anisotropy of the material examined takes into account the correlation between UTS / YS, E and porosity. Error bars for UTS and YS measurements are shown in Fig. 6, similar to error bars for elongation. The value of porosity represents the actual measurement result, not the averaged values. As suggested by the reviewer, the error bars for both UTS, YS and E were entered into the chart, but as a result, the information contained in the chart was impossible to read.

Point 16: (Line 333) SLM or DLMS? Please, clarify which approach was followed.

Response 16: The mistake has been corrected in the manuscript

Point 17: (Line 375) Please, include error bars in Fig. 9

Response 17: Hardness measurements were made on two surfaces - parallel and perpendicular to the direction of building samples. Five measurements were made on each of these surfaces. Therefore, the values shown in Fig. 9 are not averages, they are actual results obtained during the test.

Round  2

Reviewer 3 Report

 In its present state, the paper entitled “Mechanical properties and microstructure of DMLS Ti6Al4V alloy dedicated to biomedical applications” is acceptable for publication. Authors have addressed all the points highlighted by the reviewers.